# Biological Viral Infection Watermarking Architecture of MPEG/H.264/AVC/HEVC

**Bong-Joo Jang [1], Suk-Hwan Lee [2],*, Young-Suk Lee [3] and Ki-Ryong Kwon [4],***

[1] Department of Land, Water and Environment Research, KICT, Daehwa-dong, Ilsanseo-gu, Goyang-si 10223, Korea
[2] Department of Information Security, Tongmyong University, Yongdang-dong, Nam-gu, Busan 48520, Korea
[3] Institute of Image and Cultural Contents, Dongguk University, Seoul 04620, Korea
[4] Department of IT Convergence and Application Engineering, Pukyong National University, Nam-gu, Busan 608737, Korea
* Correspondence: sukhwanlee@gmail.com (S.-H.L.); kiryongkwon@gmail.com (K.-R.K.);
Tel.: +82-51-629-1285 (S.-H.L.); +82-51-629-6257 (K.-R.K.)

**Abstract:** This paper addresses the viral infectious watermarking (VIW) model using biological virus infection for a new-paradigm of video copyright protection of MPEG/H.264/AVC/HEVC. Our model aims to spread or infect the watermark to different codecs each time video contents are copied, edited, or transcoded. Thus, we regard the watermark as the infectious virus, the video content as the host, and the video codec as the contagion medium and then model pathogen, mutant, and contagion as the infectious watermark. Then, we define the techniques of viral infectious watermark generation, kernel-based VIW, and content-based VIW. Furthermore, we present a reversible VIW for fast infection in VIW model. This makes the video quality and strength be adaptively controlled in the infectious process. Experiment results verified that our VIW model can detect or recover the reversible watermark without loss in different codecs and also can maintain the quality of video content that is recovered to the same bit rate.

**Keywords:** viral infectious watermarking; video copyright protection; biological virus modeling; kernel-based watermarking; content-based watermarking

---

## 1. Introduction

As the service of ultra-definition video contents becomes more common in broadcasting, digital cinema, and home theater, many researchers have studied deep into high efficiency video coding (HEVC) and highly improved video codecs for effectively compressing and transmitting ultra-definition video contents. With HEVC, the scalable video coding (SVC) facilitates to partially decode from one sequence of compressed video to a combination of resolution, quality, and frame rate that are optimized on platform and transmission bandwidth of user device. With the marvelous progress of video codec technologies, the disputes of copyright and ownership for preventing illegal copying and distribution and pirate edition have been issued steadily as problems on commonplace video compression. To solve disputes of copyright and ownership, many nations have legislated the standardized agreement of creative commons license (CCL) that authorizes the copyright management since 2002. However, this agreement does not protect the copyright infringement all the way in cases of illegal copying and distribution of black works or partial edition. Moreover, the wide use of online storage services, torrent programs, or cloud systems makes it very difficult to guarantee ownership rights or prevent illegal copying and distribution.

Many researchers [1–14] have worked on video copyright and ownership protection based on watermarking for decades. However, most of them do not meet all the requirements of data capacity,

reliability, and video quality. Early stage works [1–6] focused on the compressed DCT domain of MPEG-2/-4, H.264 codec or the uncompressed DWT domain, considering the robustness in the face of geometric and temporal attacks. Hartung et al. [1] presented an additive spread-spectrum watermarking technique for MPEG-2 compressed video stream that embeds the watermark in the entropy coded DCT coefficients. Swanson et al. presented an object-based transparent watermarking technique [2] and also a temporal wavelet transform-based multi-resolution watermarking technique [3]. Serdean et al. [4] presented DWT-based high capacity video watermarking invariant to geometrical attacks that uses a spatial domain reference watermark. Wang et al. [5] presented a set of robust MPEG-2 video watermarking focused on geometric processing such as cropping, removal of any rows, downscaling, frame dropping, and bit-rate reduction. Zhang et al. [6] embedded the modified 2D 8-bit watermark pattern in the compressed domain to accommodate the computational constraints of H.264/AVC.

Recent stage works [7–14] have considered different codecs and the robustness to combination of commonly used attacks with the development of codecs. Asikuzzaman et al. [7] embedded the watermark into one level of the dual-tree complex wavelet transform (DT DWT) of the chrominance channel and extracted the watermark depending on the resolution of the downscaled version of the watermarked frame and the information of that frame without using the key. Fallahpour et al. [8] generated the watermark signals by the macroblock's and frame's indices and embedded them into the nonzero quantized DCT values of blocks, mostly the last nonzero values, enabling detection of spatial, temporal, and spatiotemporal tampering. Stutz et al. [9] presented a non-blind watermarking for H.264/CAVLC structure-preserving substitution with high capacity without changing the length of video stream. Khalilian et al. [10] embedded the watermark in the LL sub-band of DWT coefficients that offers the most robust PCA-based decoding. Boho et al. [11] presented an encryption-watermarking technique for H.264/AVC and HEVC by examining the practical trade-offs between the security of encryption, the robustness of watermarking, and the possibility of transcoding. Wang et al. [12] presented a real-time video watermarking that has the transparency and robustness to resist geometric distortions such as scaling, cropping, changing aspect ratio, frame dropping, and swapping.

However, most of existing techniques have difficulty in practical commercialization because of some considerable defects. The watermark embedded in a video raw data prior to compression may be lost during quantization. Furthermore, it is very difficult for the watermark to keep the robustness to geometric processing such as rotation, translation, and cropping of any frame. Watermarking techniques in the compression domain have been worked out by controlling some compression parameters to minimize the loss of watermark in the process of quantization or video editing. However, the watermark may be lost in the re-compression of decoded video data or the trans-coding by different codecs. Thus, watermarking techniques considering a specific codec have not coped with a variety of video codes [13,14] as well as standard video codecs. Furthermore, they have not coped with a multi-view video coding (MVC), which creates a variety of output streams of a single source, and a hierarchical compression, such as scalable video coding (SVC). Therefore, an effective and reliable integrated copyright protection system for video content technologies that develop and grow exponentially is needed. Besides, Niu et al. [15] presented a reversible watermarking scheme for H.264/AVC using histogram shifting of motion vector, and Xu [16] presented an efficient commutative encryption and watermarking scheme for HEVC standard, unlike H.264/AVC. Ma et al. [17] presented a video watermarking on H.264 compressed domain using the syntactic elements of the compressed bit stream, but Marren et al. [18] designed a scalable architecture for uncompressed-domain of watermarked videos using fast encoders, which re-uses the coding information from a single, previously-encoded, unwatermarked video. Abdi et al. [19] presented a real-time watermarking scheme for H.264/AVC video stream by modifying the number of nonzero-quantized AC (alternating current) coefficients in a $4 \times 4$ block of I frame.

Like living things, video codec has evolved continuously according to the IT/ICT ecosystem. Existing video watermarking systems cannot keep up with the evolution of video codecs. To solve

this problem, we consider the watermarking system and video codecs as a biological environment such as a virus. We have designed a scenario of infectious watermarking that models the relationship between video content and video codecs to biological viruses and hosts [20]. The feature of this scenario is that the watermark is continuously infected through the transcoding process of video contents through a repetitive re-embedding process on the codec. In this scenario, we introduced codec and content based watermarking techniques. The first codec-based watermarking scheme hides DCT coefficients in $4 \times 4$ block units of codec repeatedly to adapt to various codecs. The second content-based watermarking embeds the robust watermark in the ROI (Region of Interest)-based DCT coefficients of the original video data. In this scenario, however, codec-based watermarking can accumulate image degradation due to persistent and repetitive watermark re-embedding and has a limited re-embedding process on various codecs. Therefore, it is necessary to modify the scenario with a viral reversible/irreversible watermarking method for watermark authentication that is effective in the iterative detection/mutation/re-embedding process.

In this paper, we propose a scenario to integrate infectious watermark authentication, infectious watermark generation and management, content-based watermark embedding, and codec-based watermark embedding technique for continuous watermark infection through continuous transcoding detection/mutation/re-embedding. We also propose a codec-based infectious watermarking technology incorporating irreversible and reversible watermarking techniques. In this paper, we call this scenario a viral infectious watermarking (VIW) model that uses the biological virus theory for an integrated copyright protection system that copes with various video codecs such as H.264/AVC or HEVC.

In our VIW model, we assume two things. The codecs are regarded as the host with infective agent since the infected video by the first watermark, called pathogen, passes over any codecs through a number of routes. The infected video is, secondly, infected by two kinds of watermark—mutant and contagion—whenever it passes over processes of playing, streaming, editing, or transcoding. Following these assumptions, we define our VIW model by four steps: Viral infectious watermark generation and management, kernel-based VIW, content-based VIW, and VIW verification. Then, we presented total irreversible and reversible kernel-based VIW methods and content-based VIW method in our model. The existing kernel-based watermarking method [20] embeds irreversible watermark bits in high frequency coefficients in DCT $4 \times 4$ block units. The proposed kernel-based method embeds the reversible watermark in units of MB blocks according to the strength, adaptive to the inter/intra frame, and embeds the irreversible watermark in the existing high frequency coefficients in units of $4 \times 4$ blocks. Therefore, unlike the conventional method [20], the proposed method simultaneously detects both reversible and irreversible watermarks, thereby detecting the watermark adaptively in the detection/mutation/re-embedding process. From experimental results, we verified that our VIW model based on biological virus theory is effective in integrated or multiple video codecs.

This paper is organized as follows. Section 2 introduces the scenario of VIW model and related video watermarking. Section 3 describes the techniques for our VIW model. Section 4 analyzes the experimental results of our watermarking, and Section 5 then presents our conclusions of this paper.

## 2. Biological Viral Infectious Watermarking (VIW) Architecture

Previous infectious watermarking models [20] have presented a theoretical model between biological virus models and watermarking. In this chapter, we address a new VIW (Virus Infectious Watermarking) model consisting of infectious watermark generation/management, kernel-based watermarking theory for seamless watermark infection for various codecs, content based watermarking theory for copyright protection, and infectious watermark authentication.

### 2.1. Biological Virus Model

This section briefly discusses biological virus models and watermarking theories based on the previous scenarios [20]. Figure 1a shows the mapping of biological virus model and watermarking model.

- DNA/RNA: It can be considered as the watermark for copyright and access control of valuable video content.
- Capsid: It is a nucleic acid surrounded by a protective protein coat. It can be considered as the process of encryption and decryption of the copyright information.
- Virion: The combination of DNA and capsid generates virions of complete virus particle. Virions can be classified as pathogen, mutant, and contagion, depending on the way and type of infection. We use the name and concept of pathogen, mutant, and contagion and model VIW by encrypting the information of copyright and authority of videos.

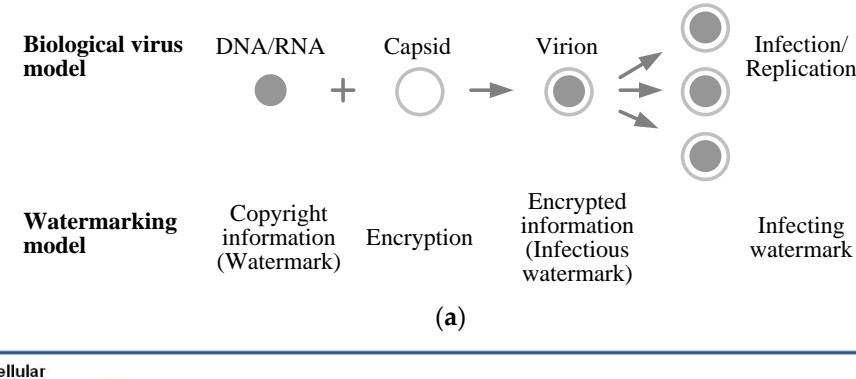

(**a**)

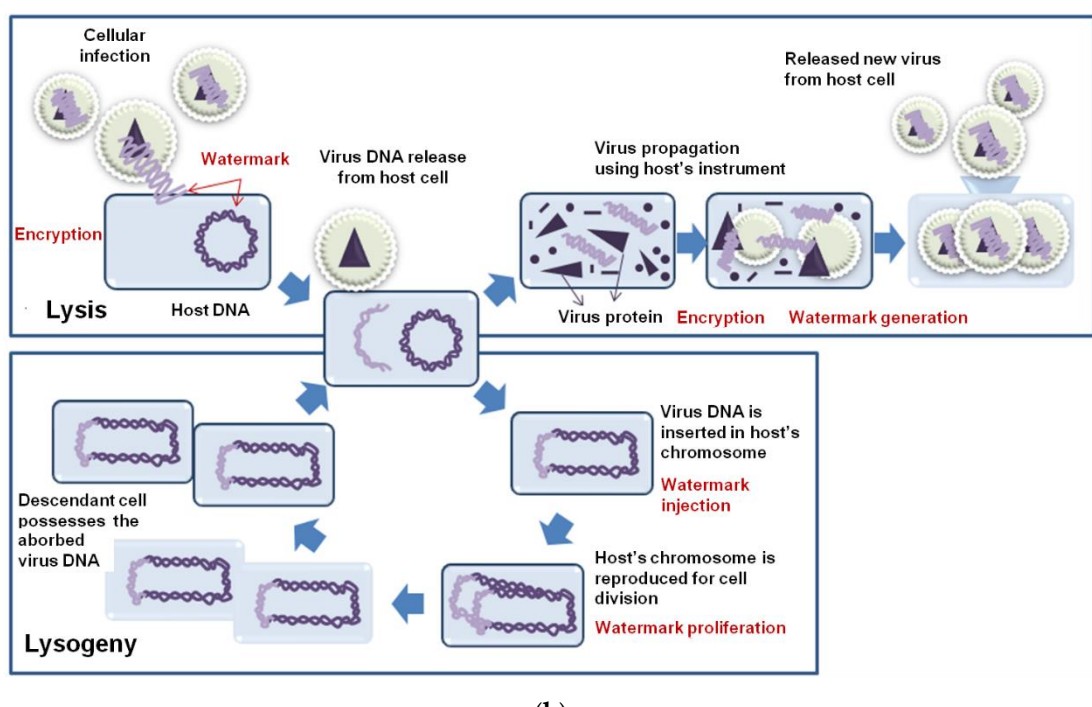

(**b**)

**Figure 1.** (**a**) Mapping of biological virus model to watermarking model and (**b**) virus infection process.

We will now consider the infection and reproduction of virus as shown in Figure 1b.

2.1.1. Virus Infection

Main properties of virus infection can be viewed in a watermarking prospect.

(1)　The infection and growth of virus have been accomplished by the process that a virion breaks in a target cell and delivers genetic material to cell. We model the watermarking process as the virus infection and growth process.

(2)　　Here, there are many ways how descendant virions are made on the type of cells. Different ways can be modeled as watermarking methods that are compatible with the types of contents or codecs.

(3)　　When the virus spreads to cells, hundreds of descendants are reproduced from the virion. This reproduction can be modelled as a re-hiding process of the infectious watermark during content reproduction or transcoding of video.

### 2.1.2. Virus Reproduction

A virus has two ways of reproduction: lysis and lysogeny.

1.　　A lysis virus that disassembles or destroys a host cell can be modeled as an authority code or permission code for decoded video contents or display devices. A lysogeny virus that hides genetic material in host cells until it becomes reproductive can be modeled as copyright protection or authentication for hidden watermarks.

2.　　Virus infections within a host cell are considered as a process of infectious watermarking. The infective ways of a virus are different on the structure of the host cell. When a host cell is considered to be a video content, the watermarking method can be applied differently depending on the type of video content or compressed bit stream.

3.　　Because biological viruses are reproduced or spread by infected hosts, we design a watermark that should spread wholly through the reproduction or trans-coding of video contents.

### 2.2. Technical VIW Concepts for Video Codec

We define four descriptive concepts for the viral infectious watermarking (VIW) model using biological virus theory as shown in Figure 2. Our VIW model can apply many different watermarking techniques for the purposes of each descriptive concept.

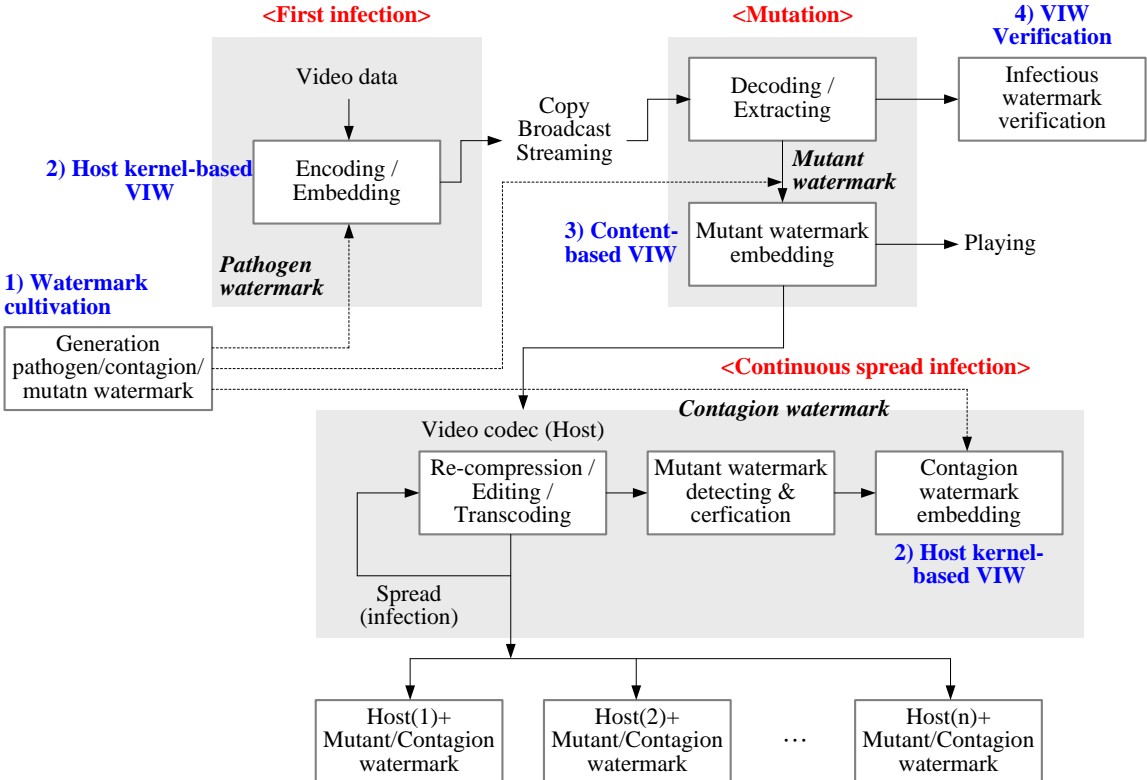

**Figure 2.** Viral infectious watermarking (VIW) model in video codec.

### 2.2.1. Watermark Cultivation for Generation and Management

This technique is to generate and manage many watermarks for copyright protection or rights/authority management of video content. Feasible works for them are as follows:

- Two watermarks on host type: Pathogen and contagion watermarks for host codec and mutant watermark for host content
- Watermark cultivation by the combination of copyright/ownership and authority: Reliable watermark cultivation by user information. Compatibility of copyright/ownership and authority in the mutation of pathogen, contagion, and mutant watermark

We define three types of watermarks in VIW model as follows:

1. Pathogen watermark: It is the infectious information for producers or copyright owners in the meaning of lysis virus. This watermark can be infected in macro-blocks of base layer in hierarchical video codec like SVC and MVC.
2. Mutant watermark: It is the mutant of pathogen watermark and it can be used in content-based VIW for decoded and reconstructed video. This watermark can be infected into all layers in hierarchical video codec.
3. Contagion watermark: It is the transfectant of extracted mutant watermark in decoding process. This watermark can be infected in the process of storing or re-compressing after video editing.

The pathogen watermark and the contagion watermark should be distinguished according to the infectious process. Two capsids of pathogen and contagion are different in the infection of biological virus. Thus, the bit-streams of pathogen and contagion watermarks are different on encryption algorithms.

### 2.2.2. Host Kernel-Based VIW

Host kernel-based VIW embeds (or injects) the watermark into host video stream for various video codecs and also infects again the watermark in decoded video data that is considered as a mutant in re-compression or trans-coding of video data. Feasible works for host kernel-based VIW are as follows:

- Infectious watermarking based on quantization, DCT/DWT kernel, or ME/MC in the encoding process.
- Reversible infectious watermarking in certification of authority and irreversible infectious watermarking in non-certification of authority.

Host-based VIW for injecting pathogen and contagion should be simple and applied to different video codecs.

### 2.2.3. Content-Based VIW

Content-based VIW infects the watermark in image data of played video stream that is the primitive video data or decoded video data. Feasible works for content-based VIW are as follows:

- Generation of first host watermarked video stream by the pathogen watermark or infected video stream by the mutant watermark;
- Robust infectious watermarking in spatial or frequency domain;
- Recovery or regeneration of attacked watermark;
- Intentional video degradation in case of non-certification.

Content-based VIW should consider spatial resolutions of trans-coding by H.264SE/MPEG-4 SVC codecs.

2.2.4. VIW Verification (WV)

VIW verification checks whether the video stream is infected with the watermark and identifies the ownership copyright or the authority from the extracted watermark. The following are related works for VIW verification.

- Reliable verification technique;
- Policy of authority control for copying, link, editing using verified infectious watermark: Authorization technique in encoder/decoder, protocol, or syntax for reliability verification and authority control.

## 3. Proposed Viral Infectious Watermarking Model

### 3.1. Overall of VIW Application for Video Codec

It is expected to be further improved by reinterpreting existing watermarking techniques and applying them to our VIW (viral infectious watermarking) model. In this sense, we design the scenario to verify the practicality of our model and examine in simulations whether primitive watermarks can be completely infected to many different video contents. In this paper, we consider H.264/MPEG-4 AVC and HEVC codecs as a primitive host and spread the watermark through H.264/MPEG-2 codecs as contagion. Figure 3 shows the VIW application for video codec by the four descriptive concepts in Figure 2.

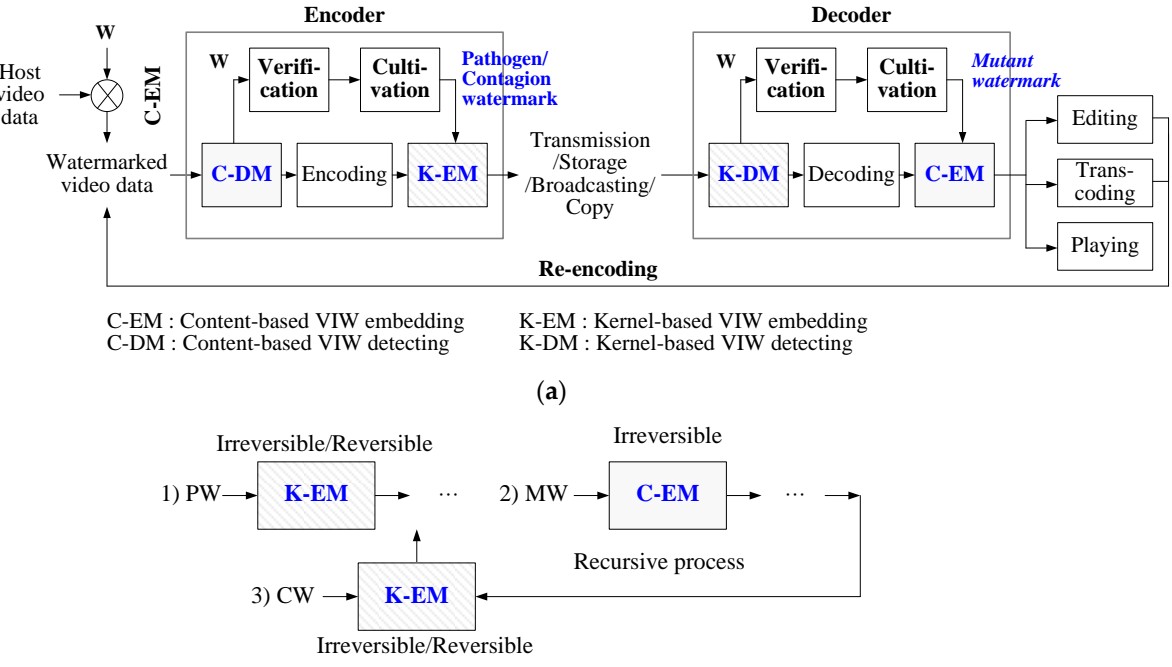

**Figure 3.** (**a**) Scenario for video watermarking in proposed VIW model and (**b**) infecting process for three watermarks.

Our algorithm in VIW model has the infectious watermark cultivation (IWC), content-based watermark embedding and detecting module (C-EM and C-DM) for the mutant watermark, irreversible/reversible kernel-based watermark embedding and detecting module (K-EM and K-DM) for the pathogen and contagion watermarks, as shown in Figure 4. We consider that the copyrighter or owner infects the watermark into a host video. The watermark infection process in encoder is as follows:

1. Before encoding or re-encoding, C-DM detects the watermark in host or re-encoded video.
2. IWC regenerates the watermark after verifying it.
3. K-EM embeds pathogen and contagion watermarks into video stream. K-EM/K-DM has two properties of irreversible and reversible. The irreversible K-EM infects the watermark wholly under trans-coding between heterogeneous codecs and guarantees the safety of video contents against editing, trans-coding, or re-compression. The reversible K-EM regenerates and authenticates the watermark continuously and controls video contents.
4. When watermarked video streams are played or edited on the decoder stage, the watermark verification and regeneration are performed in a similar process on the encoder stage. Then K-DM detects the pathogen or contagion watermark, and C-EM re-embeds the regenerated mutant watermark.

Our VIW preserves the watermark by the recursive and update process that all of watermarks and side information are repeatedly regenerated and embedded in different codes.

### 3.2. Infectious Watermark Cultivation (IWC)

Pathogen watermark $\mathbf{w}^P$, contagion watermark $\mathbf{w}^c$, and mutant watermark $\mathbf{w}^m$ are considered similar in their low complexity. The infectious information $II$ for watermark includes the copyright or ownership information, the seed number ($SN$) for video authentication, the control codes ($cc$) for parameters required to C-EM and K-EM, and the expiration date. Here we can check the system time of decoder using the expiration date of $II$ in intra-frames and then can degrade intentionally the quality of intra-frames. The infectious information for watermark $II$ for intra-frame and inter-frame can be defined by as follows:

$$II = \begin{cases} (SN(0), t_e, C_A(M, K_A), cc), & \text{if I frame} \\ (\ SN(f\%gop), C_A(M, K_A), cc), & \text{if P or B frame} \end{cases}. \tag{1}$$

Here, $SN(n)$ is the seed number for a variable $n$. A variable $n$ is 0 if the current frame is an intra-frame and then the seed number is $SN(0)$. Or a variable $n$ is $f\%gop$ if the current frame is an inter-frame and then the seed number is $SN(f\%gop)$ that depends on the group of picture $gop$ and the frame number $f$. This means that if any frame is dropped during decoding, this codec will not be authenticated. Next, $t_e$ is the expiration date on the system time. $M$ is the information of copyright or ownership and is encrypted by a symmetry-key encryption module $C_A(M, K_A)$ to prevent exposure in the watermark infectious process. If a dispute happens, the owner shall be certified the rights to the disputed video from the key $K_A$. $cc$ is the memory size of parameters that control the embedding strength on C-EM or K-EM. The length of $M$ and cc are varied while the watermark capacity is varied on different embedding algorithms or video resolutions.

The watermark $\mathbf{w}$ is the encrypted version of $II$ by the symmetry-key encryption module $C_V(\cdot)$.

$$\mathbf{w} = \mathbf{w}^P = \mathbf{w}^C = \mathbf{w}^M = C_V(II, K_V) \tag{2}$$

$K_V$ is the symmetry key for the encoder and decoder. To control the expiration date by K-EM and D-EM, we generate a key $k_C$ by the seed number of relative frames $SN(i)$, the expiration date $t_e$, and the current system time $t_s$ of that codec.

$$k_C = (-1)^u SN(i), 0 \le i < gop \tag{3}$$

$$u = \begin{cases} 0, & \text{if } i = 0 \text{ or } t_e - t_s > 0 \\ 1, & \text{otherwise} \end{cases} \tag{4}$$

$k_C$ determines the positions of watermarks.

### 3.3. Kernel-Based Watermarking (K-EM and K-DM) for Pathogen or Contagion Watermark

Our existing kernel-based watermarking [20] embeds the irreversible watermark only within intra frames, which accumulate the image degradation because of the continuous re-embedding process. However, we improve the kernel-based watermarking that has two properties of irreversible and reversible considering intra-inter-frames. Our method uses the quantized DCT coefficients that are computed at the previous stage of entropy coding in general encoding processes with a series of stages of spatial/temporal prediction, DCT transformation, quantization, scanning, and entropy coding.

#### 3.3.1. Irreversible K-EM/K-DM

(1) K-EM: Our irreversible K-EM/K-DM uses only parameters in the encoder for application of various video codecs. At first, we define the number $R^W$ of watermark bits, which are periodically infected on the frame frequency $r$, the bit number of pathogen/contagion watermark $|\mathbf{w}|$, and the numbers $N_B$ of DCT blocks in a frame.

$$R^W = |\mathbf{w}| \times \left\lfloor \frac{N_B}{|\mathbf{w}|} \times r_{\text{intra}} \right\rfloor \text{ for } \mathbf{w} = \mathbf{w}^P, \mathbf{w}^C \tag{5}$$

Hereby, we set $r = 1$ and $|\mathbf{w}| = 64$ bits. Then, we compute the block complexity $C_n$ of $4 \times 4$ blocks in all intra-frame and inter-frame by DCT coefficients in high frequency region $\Omega$ except for four low frequency coefficients including DC coefficient.

$$C_n = \sum_{u=0}^{3} \sum_{v=0}^{3} (f_n(u,v))^2 - (f_n(0,0))^2 \text{ for all } n \in N_B \tag{6}$$

$f_n(u,v)$ is a DCT coefficient on $(u,v)$ position. Then, we compute the average block complexity $\overline{C}^{\text{intra}}$ on intra-frame and also the average block complexity $\overline{C}^{\text{inter}}$ on inter-frame separately.

We arrange all DCT blocks in descending order of the block complexity $C_n$ and select $N_B^W$ ($N_B^W < N_B$) DCT blocks from the highest. As denoted, the rank of block complexity is $rank(C_n)$ and the block index of $r$th ranked complexity $r = rank(C_k)$ where $k = \text{rank}^{-1}(r)$, the indices of the embedding blocks are

$$\mathbf{K} = \left[ k_1, k_2, \cdots, k_{N_B^W} \right\} \text{ where } k_i = \text{rank}^{-1}(r_i) \text{ and } r_i \left( = rank(C_{r_i}) \right) < r_{i+1} \left( = rank(C_{r_{i+1}}) \right). \tag{7}$$

We select the highest coefficient $f_k^{\max}(u,v)$ in each of the embedding blocks and modify it by a watermark bit.

$$f_k'(u,v) = \overline{C}_k(w_k + \alpha_k) \text{ where } \overline{C}_k = \frac{C_k}{N_k} \text{ for all } k \in \mathbf{K} \tag{8}$$

$N_k$ is the number of high frequency coefficients $\Omega$ except for four low frequency coefficients including DC coefficient. Thus, $N_k$ is 10 in $4 \times 4$ blocks. $w_k$ is a watermark bit for $k$th embedding block. $\alpha_k$ is a parameter that controls the transparency and strength for each frame. Thus, it is defined by the difference energy of the highest coefficient $f_k^{\max}(u,v)$ and other coefficients $f_k(u,v)$.

$$\alpha_k = \frac{E_k}{N_B^W} = \frac{1}{N_B^W} \sum_{k=1}^{N_B^W} \sum_{k=1}^{} (\sum_{u=0}^{3} \sum_{v=0}^{3} (f_k^{\max}(u,v) - f_k(u,v))^2 - \left( f_k^{\max}(u,v) - f_n(0,0) \right)^2) \tag{9}$$

Considering security, we generate the key values $e_k = (e_k(u), e_k(v))$ for horizontal and vertical permutations for all blocks and permute an embedded coefficient $f_k'(u,v)$ with any coefficient in a block through key values.

$$f_k'(u,v) \leftrightarrow f_k'(e_k(u), e_k(v)), \ 0 < e_k(u), e_k(v) < 4 \tag{10}$$

(2) K-DM: In the extracting process, we detect the watermark bit using permutation key values $e_k$ and the threshold values $th_k$ as follows:

$$\hat{w}_k = \begin{cases} 1, & \text{if } \hat{f}_k(e_k(u), e_k(v)) \geq th_k \\ 0, & \text{otherwise} \end{cases} \quad \text{for all } k \in \mathbf{K} \tag{11}$$

We determine the threshold value $th_k$ by the block complexity $\hat{C}_k$ and the number $N_k$ of high frequency coefficients $\Omega$ from Equation (8).

$$th_k = \frac{\hat{C}_k}{N_k} \tag{12}$$

### 3.3.2. Reversible K-EM/K-DM

(1) K-EM: The reversible K-EM embeds bits of the watermark into DCT blocks within the MB(macro-block) for real-time processing. We define the embeddable blocks $B_m^W$ in $m$th MB as follows:

$$B_{m,n}^W = \begin{cases} B_{m,j}, z_{i,B_{m,j}} \neq 0 \\ NULL, z_{i,B_{m,j}} = 0 \end{cases}, \text{ for } 4 \leq i < 2^d \tag{13}$$

where $d = \begin{cases} 6, if\ |B| = 4 \\ 4, if\ |B| = 16 \end{cases}$ and $\begin{cases} 0 \leq j < \frac{|B|}{2}, if\ seed(R(k_C) + m) = 1 \\ \frac{|B|}{2} \leq j < |MB|, if\ seed(R(k_C) + m) = 0 \end{cases}$.

Here, the unique key $k_C$ that is generated from encoder and decoder in Equation (3) enables to detect the watermark, to recover the original video, and to authenticate the codec in VIW model. We determine the embeddable blocks $B_m^W$ using the 1-bit seed value obtained by the index $m$ of MB and the random number $R(k_C)$ and also determine the recoverable blocks $B_m^S$ corresponding $B_m^W$ as follows:

$$B_{m,n}^S = \begin{cases} B_{m,|B|-j-1}, z_{i,B_{m,|B|-j-1}} \neq 0 \\ NULL, z_{i,B_{m,|B|-j-1}} = 0 \end{cases}, \text{ for } 4 \leq i < 2^d \tag{14}$$

We finally determine target blocks $\boldsymbol{B}_{m,n}^T$ in MBs and embed the watermark and side information of $\left|\boldsymbol{B}_m^T\right|$ bits in target blocks $\boldsymbol{B}_{m,n}^T$.

$$\boldsymbol{B}_m^T = \left\{ \left( B_m^W, B_m^S \right) \middle| \forall B_{m,n}^W, B_{m,n}^W \neq NULL \right\} \tag{15}$$

For embedding $\left|\boldsymbol{B}_m^T\right|$ bits, we scan DCT coefficients $z$ in luminance target blocks $\boldsymbol{B}_m^T$ and then select first NZC (non-zero coefficient) from fourth scanned coefficient.

$$f_m^W = First\ NZC\ of\ z_{B_m^W}(j) \text{ and } f_m^S = First\ NZC\ of\ z_{B_m^S}(j) \text{ for } 4 \leq i < 2^d \tag{16}$$

The reason is that codecs that cannot apply K-EM or do not correspond to $k_C$ enable to decode only low frequency coefficients.

Given a bit $w_i$ of watermark $\mathbf{w} = \mathbf{w}^P, \mathbf{w}^C$, we embed $w_i$ using the sign of first NZC $f_i^W$ in embeddable blocks.

$$f_i^{*W} = (-1)^{x_i+1} \left| f_i^W \right| \tag{17}$$

$f_i^{*W}$ should be recovered to original value $f_i^W$ for the reversible property.

The side information $s_i$ is necessary for this reversible property. If $f_i^W$ is changed, $s_i$ is 1. Or not, $s_i$ is 0.

$$s_i = \begin{cases} 1, if\ f_i^W = f_i^{*W} \\ 0, if\ f_i^W \neq f_i^{*W} \end{cases} \tag{18}$$

We embed $s_i$ using $f_i^S$ in recoverable blocks in Equation (16).

$$f_i^{*S} = \left(f_i^S \ll 1\right) + s_i \tag{19}$$

$f_i^S$ corresponds to $f_i^W$. $s_i$ recovers $f_i^{*W}$ to $f_i^W$ by autonomous detection.

(2) K-DM: First, we decode a received video stream and find coefficients into which the watermark and side information are embedded using $k_C$ and $R(k_C)$. Then, we detect the side information $\hat{s}_i$ using LSB of entropy decoded coefficients $\hat{f}_i^{*S}$ in recoverable blocks.

$$\hat{s}_i = \begin{cases} 1, & if \;\; LSB\;of \;\; \hat{f}_i^{*S} = 1 \\ 0, & if \;\; LSB\;of \;\; \hat{f}_i^{*S} = 0 \end{cases} \tag{20}$$

$\hat{f}_i^{*S}$ can be recovered to $f_i^S$ by the reverse process of Equation (18). Denoted the recovered coefficient as $\hat{f}_i^S$.

$$\hat{f}_i^S = \hat{f}_i^{*S} \gg 1 = f_i^S \tag{21}$$

We detect a bit of watermark from a coefficient $\hat{f}_i^{*W}$ in embeddable blocks

$$\hat{w}_i = \begin{cases} 1, & if \;\; \hat{f}_i^{*W} > 0 \\ 0, & if \;\; \hat{f}_i^{*W} < 0 \end{cases} \tag{22}$$

and then recover $\hat{f}_i^W$ from $\hat{f}_i^{*W}$ using $\hat{s}_i$.

$$\hat{f}_i^W = (-1)^{\hat{s}_i + 1} \hat{f}_i^{*W} \tag{23}$$

### 3.4. Content-Based Watermarking (C-EM and C-DM) for Mutant Watermark

We design content-based infectious watermarking based on SVC watermarking method [8] and APDCBT (all phase discrete cosine biorthogonal transform) [21,22] that robust to H.264SE/MPEG-4 SVC/HEVC as well as different video codecs or trans-coding. Compared to DCT transform, APDCBT transform has excellent performance for high frequency attenuation and low frequency aggregation [21, 22]. Our method embeds the first watermark into spatial base layers and embeds the second watermark into spatial enhancement layers using interpolation and quantization table.

To infect the mutant information, we determine a region of interest (ROI) on the spatial resolution of each frame $f(x, y)$;

$$R(i, j) \in f(x, y), \; a \leq i \leq (a + R_H), \; b \leq j \leq (b + R_V). \tag{24}$$

$R_H$ and $R_V$ are the horizontal and vertical sizes of ROI that are determined by the scaling parameter $\gamma$, $0 < \gamma \leq 1$. $a$ and $b$ are the reference coordinate values of ROI. And we transform ROI region $R(i, j)$ to APDCBT domain in each frame and obtain the transform coefficient $\mathbf{Y} = [y(u, v)]$.

$$\mathbf{Y} = \mathbf{B}\mathbf{R}\mathbf{B}^T \tag{25}$$

The APDCBT transform matrix $\mathbf{B}$ with $R_H \times R_V$ dimensions is defined as follows:

$$B(u, v) = \begin{cases} \frac{R_H - u}{R_H R_V}, & u \in [0, R_H], v = 0 \\ \frac{1}{R_H R_V}\left((R_H - u)\cos\left(\frac{uv\pi}{R_H}\right) - \csc\left(\frac{v\pi}{R_V}\right)\cos\left(\frac{uv\pi}{R_H}\right)\right), & u \in [0, R_H], v \in [0, R_V] \end{cases} \tag{26}$$

We extend the quantization table $\mathbf{Q} = \{Q(m,n)/2\}$ $(0 \leq m, n < 8)$ for intra frame in H.264 to the horizontal and vertical size of ROI using bilinear interpolation.

$$Q_{\text{ROI}}(u,v) = \begin{cases} Q(m,n), & \text{if } u = m \times l_V \text{ and } v = n \times l_H \\ \left\lfloor \sum\limits_{k=-1}^{1} \sum\limits_{l=-1}^{1} \omega(k,l)Q(u+k,v+l) \right\rfloor, & \text{if } u = \left\lfloor \frac{2m+1}{2} \times l_V \right\rfloor \text{ and } v = \left\lfloor \frac{2n+1}{2} \times l_H \right\rfloor \\ \vdots & \vdots \\ Q(u,v), & \text{if } u = \tau \times \frac{R_V}{8 \times 2^n} - 1 \text{ and } v = \tau \times \frac{R_H}{8 \times 2^n} - 1 \end{cases} \tag{27}$$

$l_V$ and $l_H$ are $l_V = \left\lceil \tau \times \frac{R_V}{8 \times 2^n} \right\rceil$ and $l_H = \left\lceil \tau \times \frac{R_H}{8 \times 2^n} \right\rceil$. $\boldsymbol{\omega} = \{\omega(k,l)\}$ is $3 \times 3$ weighting random matrix that are generated by normal distribution.

$$\boldsymbol{\omega} = \frac{1}{\sum_{k=0}^{2} \sum_{l=0}^{2} \omega(k,l)} \mathbf{GR} \tag{28}$$

where

$$\mathbf{R} = \begin{bmatrix} r(t_{0,0}) & r(t_{0,1}) & r(t_{0,2}) \\ r(t_{1,0}) & r(t_{1,1}) & r(t_{1,2}) \\ r(t_{2,1}) & r(t_{2,1}) & r(t_{2,2}) \end{bmatrix} \text{ and } \mathbf{G} = \frac{1}{16} \begin{bmatrix} 1 & 2 & 1 \\ 2 & 4 & 2 \\ 1 & 2 & 1 \end{bmatrix} \tag{29}$$

$r(t_{k,l})$ is random value that is generated by normal distribution of mean = 0.5 and variance = 1 with a seed $t_{k,l}$.

The extended quantization table $Q_{\text{ROI}}(u,v)$ is used as a watermark key. We align APDCBT coefficients $c(u,v)$ of ROI region by zig-zag scan ordering and quantize them by $Q_{\text{ROI}}(u,v)$. Then, we select 64 coefficients in low frequency region and embed the watermark bits into specific bits of quantized APDCBT coefficients as follows:

$$c'(u,v) = \begin{cases} \left\lfloor \frac{c(u,v)}{\alpha \times Q_{ROI}(u,v)} \right\rfloor \| (1 \ll \delta) & \text{if } w = 1 \\ \left\lfloor \frac{c(u,v)}{\alpha \times Q_{ROI}(u,v)} \right\rfloor \&\& \sim (1 \ll \delta) & \text{if } w = 0 \end{cases} \tag{30}$$

$\|$ and $\&\&$ are bitwise OR and bitwise AND operators. $\sim$ is a bitwise inverse operator. The quantization scale factor $\alpha$ and the bit-shift $\delta$ are determined by considering the embedding strength and invisibility. We obtain the embedded ROI $R\prime(i,j)$ through inverse quantization and inverse APDCBT and combine it with non-ROI.

The extracting process is performed using a watermark key $Q_{\text{ROI}}(u,v)$ similar to the above embedding process. Thus, the bits of watermark in each frame can be detected by checking $\delta$th bits of quantized APDCBT coefficients $\left\lfloor \frac{\widetilde{c}(u,v)}{\alpha \times Q_{ROI}(u,v)} \right\rfloor$ as follows:

$$\widetilde{w} = \begin{cases} 1, & \text{if } \left( \left\lfloor \frac{\widetilde{c}(u,v)}{\alpha \times Q_{ROI}(u,v)} \right\rfloor \&\& (1 \ll \delta) \right) == 1 \\ 0, & \text{otherwise} \end{cases} \tag{31}$$

## 4. Experimental Results

### 4.1. Experimental Environment

We examined from simulations whether the primitive watermark could be entirely infected to different video codecs. Our simulation considered H.264/MPEG-4 AVC codecs as a primitive host and embedded or spread the infectious watermark by H.264/MPEG-4 AVC, MPEG-2, HEVC (H.265) codecs as contagion and simulated all codecs by FFmpeg; MPEG-2 Part2 (mpeg2video), H.264/MPEG-4 AVC (libx264), HEVC (libx265). Moreover, our simulation was implemented by C/C++ in the PC with Intel

Core i7-2700K CPU@3.50GHz, 4cores/8threads and 32GB RAM, of which the OS is 64-bit Windows 10 Ultimate K.

The simulation scenario for verification of our VIW model, as shown in Figure 4. We verified the infectious of watermark with two strategies. The first is that we decoded pathogenic video contents by H.264 SVC and re-coded them by MPEG-2 codec (or H.264/MPEG-4 AVC) and then verified the infectious of watermarks in re-coded MPEG-2 (or H.264/MPEG-4 AVC) streams. The second is that we trans-coded the re-coded MPEG-2 (or H.264/MPEG-4 AVC) streams by H.264/MPEG-4 AVC (or HEVC (H.265)) codec and then verified the infectious of pathogen, mutant, contagion watermarks.

We used MPEG2 and H.264/MPEG-4 AVC for infectious codec of low resolutions (CIF, 4CIF etc.) and used H.264/MPEG-4 AVC and HEVC for infectious codecs for high resolutions (720 p, 1080 p). Tables 1 and 2 show simulation environments for low and high resolutions. Here, the quality is controlled by a *qscale* parameter (0–32) in MPEG-2 and a *crf* parameter (0–51) in H.264/MPEG-4 AVC and HEVC. We set a *qscale* to 4 in MPEG-2 and a *crf* to 23 in H.264/MPEG-4 AVC and a *crf* to 28 in HEVC, which are default values in codec. The following are execution examples of FFmpeg for three codecs; MPEG-2 (ffmpeg -i input -codec:v mpeg2video -qscale:v 4 -codec:a copy output), H.264/MPEG-4 AVC (ffmpeg -i input -c:v libx264 -preset slow -crf 23 -c:a copy output.mkv), HEVC (ffmpeg -i input -c:v libx265 -crf 28 -c:a copy output).

**Table 1.** Infectious simulation scenario 1 using MPEG-2 and H.264/MPEG-4 AVC for low resolution (64-bit watermark).

| Infection Target | Encoder | | Decoder | | Output | Detection Watermark |
|---|---|---|---|---|---|---|
| | Codec | Watermark | Codec | Watermark | | |
| Host video | SVC | Pathogen PW | SVC | Mutant MW1 | PW/MW1 video | PW |
| Output 1 | MPEG2 (qscale = 8) | Contagion CW1 | MPEG2 (qscale = 8) | Mutant MW2 | PW/MW1,2/CW1 video | PW/MW1/CW1 |
| Output 2 | H.264 (crf = 23) | Contagion CW2 | H.264 (crf = 23) | Mutant MW3 | PW/MW1,2,3/CW1,2 video | PW/MW1,2/CW1,2 |

**Table 2.** Infectious simulation scenario 1 using H.264/MPEG-4 AVC and HEVC for high resolution (128-bit watermark).

| Infection Target | Encoder | | Decoder | | Output | Detection Watermark |
|---|---|---|---|---|---|---|
| | Codec | Watermark | Codec | Watermark | | |
| Host video | SVC | Pathogen PW | SVC | Mutant MW1 | PW/MW1 video | PW |
| Output 1 | H.264 (crf = 23) | Contagion CW1 | H.264 (crf = 23) | Mutant MW2 | PW/MW1,2/CW1 video | PW/MW1/CW1 |
| Output 2 | HEVC (crf = 28) | Contagion CW2 | HEVC (crf = 28) | Mutant MW3 | PW/MW1,2,3/CW1,2 video | PW/MW1,2/CW1,2 |

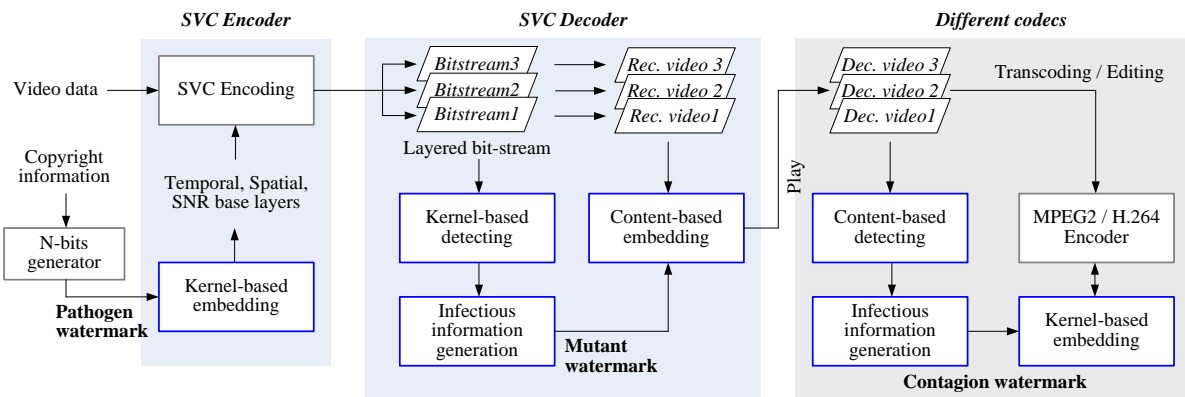

**Figure 4.** Simulation scenario of our infectious watermarking.

We used test sequences of CIF, 4CIF, and 544p for low resolutions and 720 p, 1280 p, and 2K for high resolutions and some movie sequences were downloaded in H264info.com. Figure 5 shows test video sequences with low resolution used in our experiment.

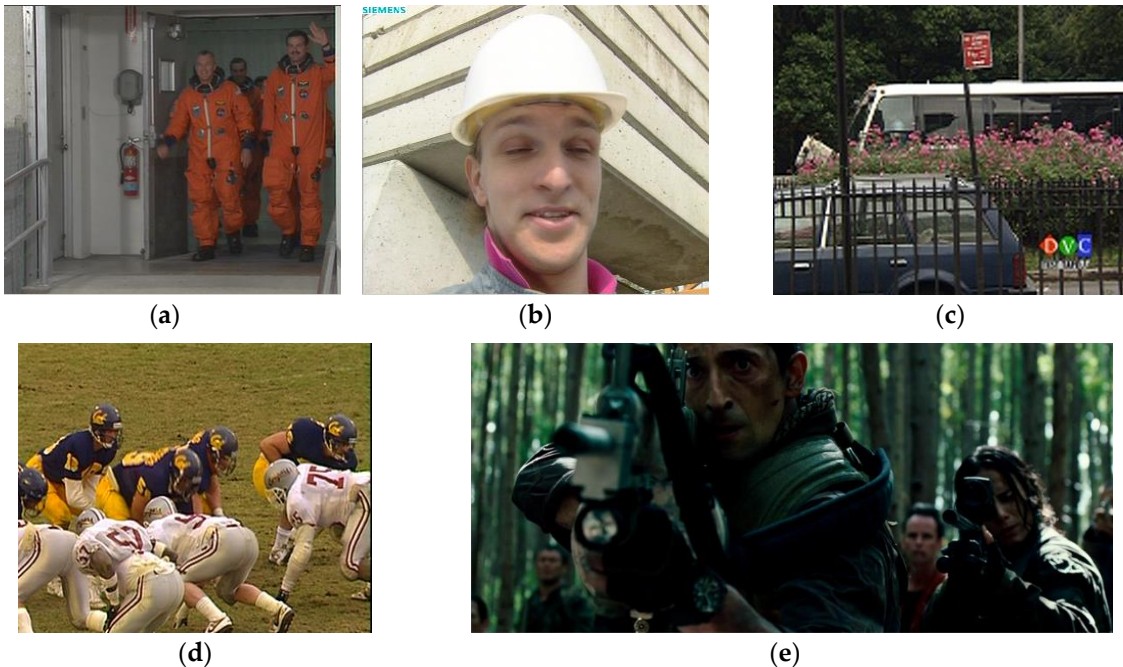

**Figure 5.** Test video sequences; (**a**) Crew (4CIF@30fps), (**b**) Foreman (CIF@30fps), (**c**) Bus (CIF@30fps), and (**d**) Football (CIF@30fps), and (**e**) movie (1280 × 544@24fps).

We compared BERs and PSNRs of our method, previous infectious watermark method [20], and mixture method [19,21] in our scenario. A mixture method is used for K-EM/K-DM of Abdi's method [19] and C-EM/C-DM of Yang's method.

*4.2. Performance Comparison of BER and PSNR*

4.2.1. BER Comparison

Our experiment applied the irreversible/reversible K-EM/DM on base layers for temporal, spatial, and SNR scalabilities and encoded different layers in SVC to bit-streams independently of each other. Since the watermark in base layer can be detected in different layers, temporal and SNR scalabilities were encoded to one layer for the low complexity.

For the fair evaluation, we repeatedly embedded the same 64-bit/128-bit watermark (pathogen, mutant, contagion) per frame for all methods. The mutant watermarks (MW = MW1, MW2, MW3) and the contagion watermark (CW = CW1, CW2) embedded on each round are used as the same watermark of MW and CW in Tables 1 and 2. We detected the infectious watermark in K-EM/DM and C-EM/DM, as shown in Tables 1 and 2 and presented the results in Tables 3 and 4.

According to these results, the proposed method and the existing viral method showed that PW, MW, and CW watermarks were detected without errors until the first infection, but bit errors of both methods occurred from the second infection. That is, the proposed method has average BERs of 0.006–0.018 for PW and MW watermarks and 0.016–0.03 for CW watermark. The conventional viral method has average BERs of 0.012–0.023 for PW and MW watermarks and 0.022–0.048 for CW watermark. BERs of PW and MW of Yang's method increased from 0.012 to 0.083 on average from compression of host data to 1st and 2nd infection. In addition, BER of CW of Abdi's method increased from 0.004 to 0.038 from the first infection to the second infection.

**Table 3.** BER results of pathogen, contagion, and mutant watermarks by MPEG2 and H.264/MPEG4 AVC for low resolutions.

| Test Sequences | Resolution@FPS | Host | | | 1st Infection | | | 2nd Infection | | |
| | | H.264 SVC | | | MPEG-2 (qscale 4) | | | H.264 (crf 23) | | |
| | | PW(C-EM/DM) BER | | | CW1(K-EM/DM) BER/PW/MW1(C-EM/DM) BER | | | CW1,2(K-EM/DM) BER/PW/MW1,2(C-EM/DM) BER | | |
| | | Propose | Previous [20] | Yang [21] | Propose | Previous [20] | Abdi [19]/Yang [20] | Propose | Previous [20] | Abdi [19]/Yang [20] |
|---|---|---|---|---|---|---|---|---|---|---|
| Football | CIF@30fps | 0 | 0 | 0.02 | 0/0 | 0/0 | 0.01/0.04 | 0.01/0.02 | 0.02/0.03 | 0.03/0.10 |
| Foreman | CIF@30fps | 0 | 0 | 0.01 | 0/0 | 0/0 | 0.00/0.02 | 0.00/0.02 | 0.01/0.02 | 0.02/0.07 |
| Bus | CIF@30fps | 0 | 0 | 0.00 | 0/0 | 0/0 | 0.00/0.01 | 0.01/0.01 | 0.01/0.01 | 0.02/0.05 |
| Crew | 4CIF@30fps | 0 | 0 | 0.01 | 0/0 | 0/0 | 0.00/0.02 | 0.00/0.01 | 0.01/0.01 | 0.03/0.08 |
| Movie | 544p@24fps | 0 | 0 | 0.02 | 0/0 | 0/0 | 0.01/0.03 | 0.01/0.02 | 0.01/0.03 | 0.04/0.10 |
| Average BER | | 0 | 0 | 0.012 | 0/0 | 0/0 | 0.004/0.024 | 0.006/0.016 | 0.012/0.022 | 0.028/0.080 |

**Table 4.** BER results of pathogen, contagion, and mutant watermarks by H.264/MPEG4 AVC and HEVC for high resolutions.

| Test Sequences | Resolution @23.967FPS | Host | | | 1st Infection | | | 2nd Infection | | |
| | | H.264 SVC | | | H.264 (crf 23) | | | HEVC (crf 28) | | |
| | | PW(C-EM/DM) BER | | | CW1(K-EM/DM) BER/PW/MW1(C-EM/DM) BER | | | CW1,2(K-EM/DM) BER/PW/MW1,2(C-EM/DM) BER | | |
| | | Propose | Previous [20] | Yang [21] | Propose | Previous | Abdi [19]/Yang [20] | Propose | Previous | Abdi [19]/Yang [20] |
|---|---|---|---|---|---|---|---|---|---|---|
| Simpsons | 1080p | 0 | 0 | 0.03 | 0/0 | 0/0 | 0.01/0.05 | 0.02/0.03 | 0.03/0.05 | 0.04/0.08 |
| IamLegend | 1080p | 0 | 0 | 0.04 | 0/0 | 0/0 | 0.02/0.06 | 0.02/0.04 | 0.02/0.06 | 0.05/0.11 |
| Serenity | 720p | 0 | 0 | 0.02 | 0/0 | 0/0 | 0.01/0.04 | 0.01/0.02 | 0.02/0.04 | 0.03/0.06 |
| Fantastic4 | 720p | 0 | 0 | 0.03 | 0/0 | 0/0 | 0.02/0.06 | 0.02/0.03 | 0.02/0.04 | 0.03/0.08 |
| Average BER | | 0 | 0 | 0.03 | 0 | 0/0 | 0.015/0.053 | 0.018/0.030 | 0.023/0.048 | 0.038/0.083 |

### 4.2.2. PSNR Comparison

Our K-EM/C-EM infects the watermark of three types irrespective of the different encoding parameters. We computed PSNRs to analyze the degraded quality by infected watermarks in various codecs in Tables 1 and 2 and presented PSNRs of different codecs in Tables 5 and 6. These PSNRS were the average of the results of 100 experiments, repeatedly. SVC trans-coding of host video sequence does not degrade the quality, which is not different subjectively or objectively. However, the first infection does slightly degrade the quality. This is because, unlike pathogenic watermarks, mutant watermarks are embedded on the basis of C-EM/DM, primarily to consider robustness. However, due to the deterioration and preservation of the infectious watermarks, no further image degradation occurred after the second infection.

**Table 5.** PSNR results of pathogen, contagion and mutant watermarks by MPEG2 and H.264/MPEG4 AVC for low resolution.

| Test Sequences | Resolution@FPS | Host | | | 1st Infection | | | 2nd Infection | | |
| | | H.264 SVC | | | MPEG-2 (qscale 4) | | | H.264 (crf 23) | | |
| | | PW(C-EM/DM) PSNR | | | CW1(K-EM/DM) PSNR/PW/MW1(C-EM/DM) PSNR | | | CW1,2(K-EM/DM) PSNR/PW/MW1,2(C-EM/DM) PSNR | | |
| | | Propose | Previous [20] | Yang [21] | Propose | Previous [20] | Abdi [19]/Yang [20] | Propose | Previous [20] | Abdi [19]/Yang [20] |
|---|---|---|---|---|---|---|---|---|---|---|
| Football | CIF@30fps | 41.39 | 41.14 | 40.73 | 36.92 | 36.46 | 36.78/35.45 | 35.66 | 36.77 | 34.27/33.63 |
| Foreman | CIF@30fps | 42.21 | 41.95 | 41.53 | 37.36 | 36.48 | 36.81/35.78 | 35.95 | 36.30 | 34.76/34.17 |
| Bus | CIF@30fps | 40.45 | 40.08 | 39.68 | 36.93 | 36.29 | 36.47/35.76 | 36.13 | 35.82 | 35.20/33.96 |
| Crew | 4CIF@30fps | 40.09 | 39.41 | 39.06 | 35.65 | 35.26 | 35.56/34.64 | 35.4 | 35.07 | 34.34/33.46 |
| Movie | 544p@24fps | 40.89 | 40.28 | 39.88 | 37.21 | 36.34 | 36.75/35.83 | 35.86 | 36.03 | 34.82/34.01 |
| Average PSNR | | 40.99 | 40.57 | 40.18 | 36.81 | 36.17 | 36.47/35.49 | 35.80 | 35.19 | 34.68/33.85 |

**Table 6.** PSNR results of pathogen, contagion, and mutant watermarks by H.264/MPEG4 AVC and HEVC for high resolution.

| Test Sequences | Resolution @23.967fps | Host | | | 1st Infection | | | 2nd Infection | | |
|---|---|---|---|---|---|---|---|---|---|---|
| | | H.264 SVC | | | H.264 (crf 23) | | | HEVC (crf 28) | | |
| | | PW(C-EM/DM) PSNR | | | CW1(K-EM/DM)/ PW/MW1(C-EM/DM) PSNR | | | CW1,2(K-EM/DM)/ PW/MW1,2(C-EM/DM) PSNR | | |
| | | Propose | Previous [20] | Yang [21] | Propose | Previous [20] | Abdi [19]/Yang [20] | Propose | Previous [20] | Abdi [19]/Yang [20] |
| Simpsons | 1080p | 40.88 | 40.44 | 39.72 | 39.25 | 38.85 | 39.15/38.06 | 38.26 | 37.39 | 36.98/36.22 |
| IamLegend | 1080p | 40.43 | 40.37 | 39.35 | 38.85 | 38.42 | 38.71/37.74 | 37.83 | 36.98 | 36.58/36.18 |
| Serenity | 720p | 41.56 | 41.07 | 40.64 | 40.21 | 39.50 | 39.78/38.69 | 38.89 | 37.92 | 37.57/37.21 |
| Fantastic4 | 720p | 41.78 | 41.62 | 40.95 | 40.42 | 39.70 | 40.04/38.97 | 39.09 | 38.23 | 38.02/37.14 |
| Average PSNR | | 41.16 | 40.88 | 40.17 | 39.68 | 39.12 | 39.42/38.37 | 38.52 | 37.63 | 37.29/36.69 |

These tables show that our method has average PSNRs of 40.99–41.16 dB by PW watermark embedded in the host data, 36.81–39.68 dB by the first infected PW, MW, CW watermarks in the first MPEG2 and H.264 transcoding process and 35.80–38.52 dB by the second infected PW, MW, CW watermarks in the second H.264 and HEVC transcoding process. Thus, we know that PSNR is reduced by 1.01–4.176 dB by transcoding and watermark embedding by the first and second infections from the host data.

The PSNR of the conventional infectious method [20] was 40.57–40.88 dB in average in host data, 36.17–39.12 dB in average in the first infection and 35.19–37.63 dB in the second infection. These average PSNR are 0.414–0.648 dB lower than average PSNR of our method. The PSNR of the combination of Abdi's [19] and Yang's [21] methods was 40.18 dB in average in host data, 35.49–39.42 dB in average in the first infection and 33.85–37.29 dB in average in the second infection. Abdi's method was Yang's method. The PSNR was slightly higher than that of the control group. The PSNR of Abdi's method was higher than the PSNR of Yang method. The PSNR of this combination method is 0.340–1.574 dB lower than the PSNR of the proposed method.

Our experiment analyzed the effects of reversible watermark according to the compression ratio of the video stream. The quality of video is degraded if the watermarking method to modify a compressed stream or quantification table meets a predefined bit rate. In addition, if the encoded data in the entropy coding stage is changed by the watermark, the bit rate is increased, which leads to high quantization parameters in the next quantization stage. Since our reversible K-EM embeds the watermark in the entropy coding step, the increase of bit rate and the quality degradation have occurred slightly. We computed the PSNRs of original sequence, recovered sequence, and degraded sequence and presented these results in Tables 5 and 6. The proposed method uses the sign value of the coefficient only for the watermark embedding, and the non-zero coefficients only for the side information embedding, thus preventing the reduction of entropy coding efficiency.

*4.3. Proposed Method Analysis*

4.3.1. Watermark Capacity

The watermark length is determined by four parameters $(SN(t), t_e, C_A(M, K_A), cc)$ for the resolution and compression ratio of the video sequence. Figure 6 shows the watermark capacity of the test sequence computed in this experiment. These results verified that the watermark can be embedded in an average 2500 MB/frames out of 5440 MB/frames. Thus, the proposed method has an average watermark capacity of 2500 bits/frames, which means that the watermark capacity is large.

According to the above results, we synthesized the infectious information $II = \{M, t_e, SN(t), cc\}$ for a watermark of a total of 512 bits in Equation (1) and generated the encrypted watermark $X$ of 512 bits by 128-bit DES block-cipher. Here, $M$ is 256 bits for copyright information, $t_e$ is 64 bits for expiration data, $SN(t)$ is 32 bits for seed number, and $cc$ is 160 bits for control memory size. Then, we computed the embeddable capacity for each frame and embedded the watermark $X$ repeatedly in accordance with the embeddable capacity.

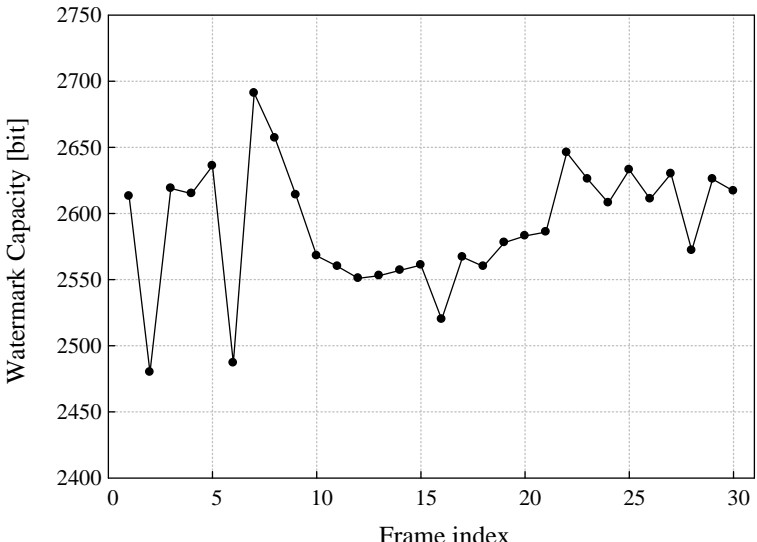

**Figure 6.** Average watermark capacity for 30 frames of test video sequence.

### 4.3.2. Attack of Compressed Stream

Our K-EM/K-DM and C-EM/DM methods detect the watermark while the sequences of video are decoded and then infect the watermark again by kernel-based reversible watermarking. Therefore, we evaluated the robustness against attacks on the compressed video stream, except for attacks on the decoded frames.

We have shown any frame in the recovered video after K-EM and K-DM processing in Figure 7a,b and also any frame in the recovered video after K-EM and K-EM processing and the re-embedding of random watermark, which is considered as attack, in Figure 7c,d.

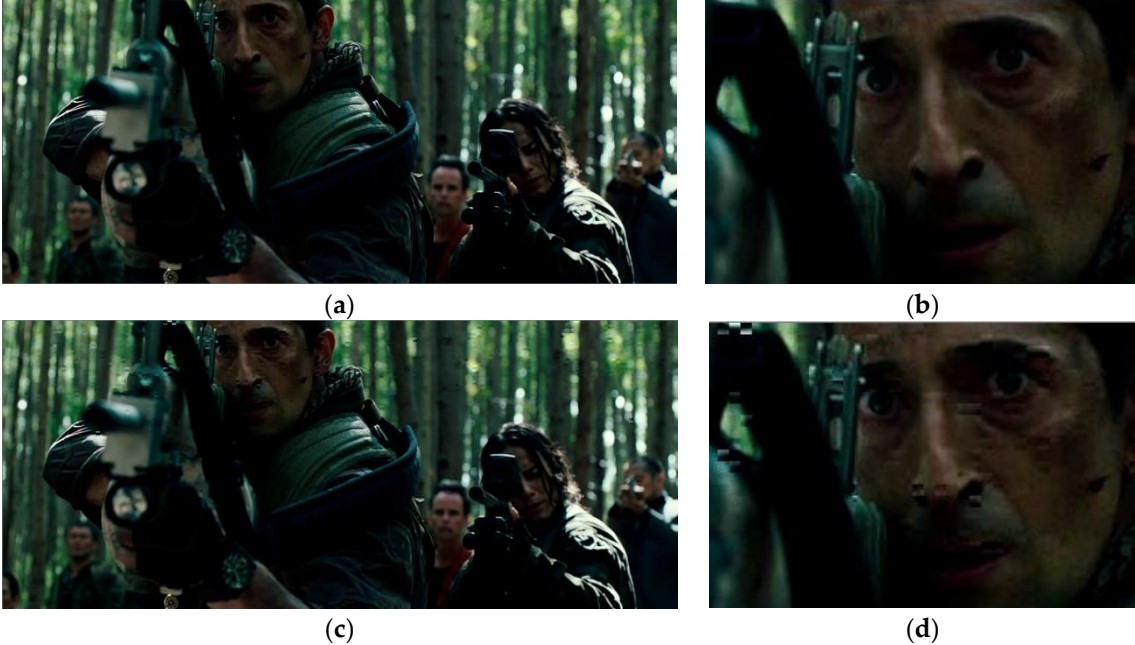

**Figure 7.** (**a**) A frame on a video sequence recovered after K-EM and K-DM processing and (**c**) a frame on a video sequence with blocking artifacts due to the recovery error by re-embedding the random watermarks. (**b**,**d**) are 30% enlarged versions of (**a**,**c**) respectively.

When a collusion attack by any watermarks or a watermark removal attack occurs in a compressed video stream, the seed number for the embedded locations and key values is damaged. Damage to the seed number of the key value will degrade the watermark and side information and result in blocking artifacts. The sporadic blocking artifacts in each frame have a detrimental effect on high-quality video streaming services. Therefore, we have confirmed that the proposed method has tolerances to attack in a compressed video stream.

### 4.3.3. Expired Video Stream

We experimented with the expired use of video stream. That is, when the system time $t_s$ of the codec expired for detecting the watermark or generating the recovery key $k_C$, we blocked the high frequency coefficient of the replay video stream containing the watermark and side information. This allows expired video streams or unauthorized codecs to be played only with low quality or low resolution.

Figure 8 shows a frame on a video stream that has been restored normally and a frame on a video stream with low quality and low resolution in an expired or unlicensed codec. We can see that the quality and resolution of expired video stream or unauthorized codec are very low.

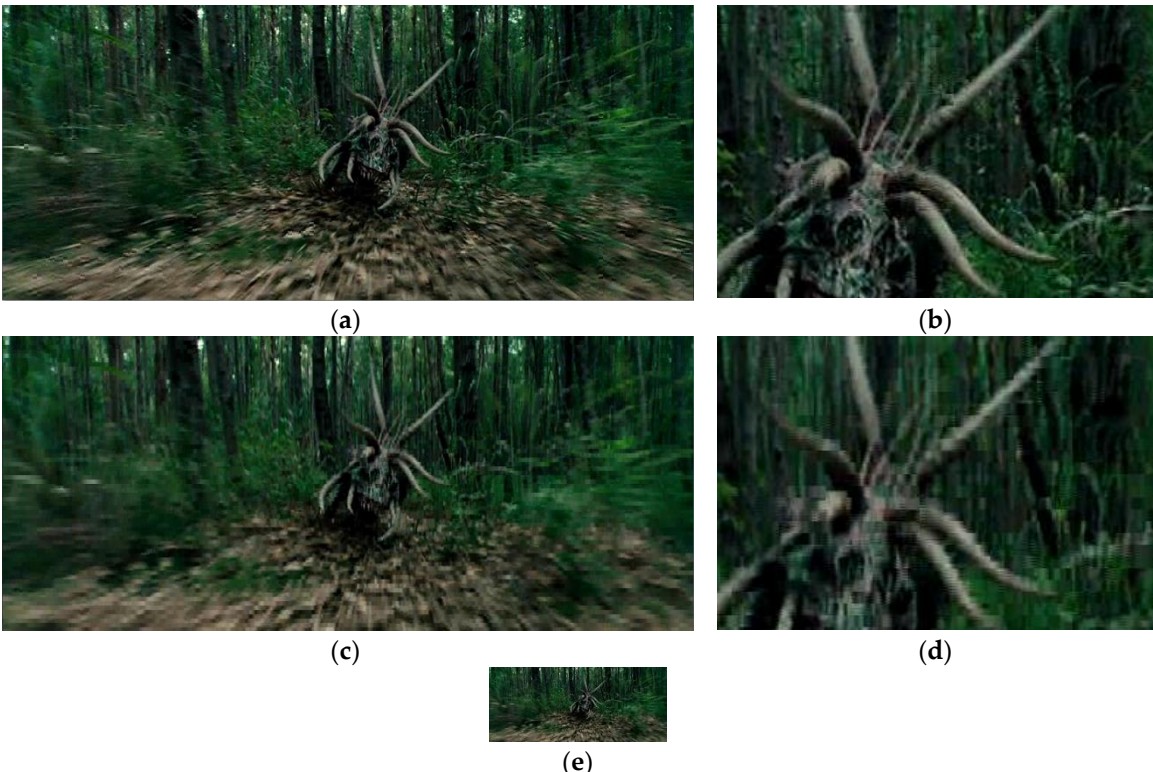

(**a**)

(**b**)

(**c**)

(**d**)

(**e**)

**Figure 8.** (**a**) A frame on a video sequence recovered after K-EM and K-DM processing, (**c**) a frame on an expired video sequence with low quality, and (**e**) a frame on an expired video sequence with both low quality and low resolution. (**b**,**d**) are 30% enlarged versions of (**a**,**c**) respectively.

## 5. Conclusions

We presented an infectious reversible watermarking that quickly and effectively infects the reversible watermark in the IIH (infectious information hiding) system for safe dissemination of video content. The proposed method can control video quality and insertion strength together by the control code for infecting the watermark in the C-EM algorithm and the watermark key that is used with the expiration date of the video content. The proposed method hides the watermark into quantized coefficients on a MB unit for real-time processing and also hides the side information for the recovery

of the original video content. Thus, the proposed method has the advantage of very low calculations of embedding processes and no frame delay to recover the original video content. Since the quality of video can be reduced by damage to the recovery information when a compressed video stream is attacked, the proposed method can prevent attempts to attack the video content. In a non-attacked video stream, there is no loss of the watermark after reversible watermark detection and video recovery. The quality of the recovered video is similar to the quality of the compressed video that has the same bit rate as this video. If the non-authorized codecs are used to decode watermarked video, or if the use of such video has expired, the proposed method provides video with a low resolution and a quality of less than 30 dB by removing the reversible watermark and performing the de-blocking. That is, the proposed method enables the application of content promotion (PR), free playback services, and other applications. We looked at the structure of infection information and error correction in IIH system, and improved the performance of content-based reversible watermarking by using control code with measuring beam and reversible watermark.

**Author Contributions:** Conceptualization: B.-J.J., S.-H.L.; Data curation: B.-J.J., S.-H.L.; Formal Analysis: B.-J.J., S.-H.L.; Funding Acquisition: K.-R.K., S.-H.L.; Investigation: B.-J.J., S.-H.L., Y.-S.L.; Methodology: B.-J.J., S.-H.L., K.-R.K., Y.-S.L.; Project Administration: K.-R.K., S.-H.L.; Resources: B.-J.J.; Softwares: B.-J.J.; Supervision: K.-R.K., S.-H.L.; Validation: K.-R.K., S.-H.L., Y.-S.L.; Visualization: B.-J.J., S.-H.L.; Writing-Original Draft Preparation: B.-J.J., S.-H.L.; Writing-Review and Editing: S.-H.L., Y.-S.L., K.-R.K.

**Funding:** This research was supported by the Basic Science Research Program through the National Research Foundation of Korea (NRF) funded by the Ministry of Science and ICT (No. 2016R1D1A3B03931003, 2017R1A2B2012456).

**Acknowledgments:** This research is technically supported by grants from Strategic Research Project (Development of Digital Pairing Core Technology for Water Resource Utilization and Disaster Prediction) funded by Korea Institute of Civil Engineering and Building Technology and also by Basic Science Research Program through the National Research Foundation of Korea (NRF) funded by the Ministry of Education, Science and Technology (2018R1D1A1B07051261).

**Conflicts of Interest:** The authors declare no conflict of interest.

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
