# Peer review of "Biological Viral Infection Watermarking Architecture of MPEG/H.264/AVC/HEVC"

_electronics, doi:10.3390/electronics8080889_

Round 1
Reviewer 1 Report
In this paper a video watermarking methodology based on biological viral infection is proposed.
After studying the manuscript and the related references the following comments are stated:
1. The manuscript has significant overlaps with authors' previous work ref[20]. Please, the authos to provide more information regarding the new contribution of this work in comparison to ref.[20].
2. Please the authors to provide more implementation and experimantation details e.g. programming environment, properties of videos, properties of computer executing the experiments,etc.
3. The main weakness of the manuscript is the absence of any comparison with other watermarking schemes. Please the authors to conduct a comparative study with other algorithms from the literature and to provide
quantitative and qualitative results.
The reviewer suggests the majoe revision of this work based on the above comments.
Author Response
The manuscript has significant overlaps with authors' previous work ref[20]. Please, the authors to provide more information regarding the new contribution of this work in comparison to ref.[20].
Answer : Our manuscript is based on the viral watermarking scenario model presented in the previous paper, and it is intended to improve this model with the viral watermark generation and management, host kernel-based VIW, content-based VIW and VIW authentication. Furthermore, we present a newly irreversible/reversible kernel-based watermarking and content-based watermarking in this manuscript.
We compared BER and PSNR of the previous work [20] and our method in Chapter 4. Experimental Results.
Revision Part : Blue-color
Section 1. Introduction (5, 6, 7 paragraphs noted in blue color) : We provide more information for new contribution of this work compared to the previous work [20]. Section 2 (1 paragraph) : We summary a new contribution compared the previous model. Subsection 3.3 (1 paragraph) : We note the difference between our kernel-based watermarking method and previous method. Subsection 3.3.3 : Reversible K-EM/K-DM is a newly method compared to the previous work. Subsection 3.4 : (1 paragraph) : We note the difference between our content-based watermarking method and previous method. Subsection 4.2 : We compared BER and PSNR of our method and previous method [20].
Please the authors to provide more implementation and experimantation details e.g. programming environment, properties of videos, properties of computer executing the experiments,etc.
Answer : According to comment, we present more information of our experiments in Subsection 4.1. This section includes our simulation scenario, test codecs/sequences, computing environment.
The main weakness of the manuscript is the absence of any comparison with other watermarking schemes. Please the authors to conduct a comparative study with other algorithms from the literature and to provide quantitative and qualitative results.
Answer : We compared BER and PSNR of proposed method, our previous work [20] and combination of Abdi’s method [19] ad Yang’s method [21] and presented the results and analysis in Subsection 4.2. We explain these conventional methods in Section 1. Introduction and Subsection 3.4.
The proposed virus watermarking model requires three watermarking methods (irreversible/reversible kernel-based watermarking methods and irreversible content-based watermarking methods). For the performance comparison, we need the combination of conventional kernel-based watermarking and content-based watermarking methods. Our experiments combined Abdi’s method for kernel-based watermarking and Yang’s method for content-based watermarking.
We colored the revised parts blue.

Reviewer 2 Report
In this manuscript, authors address a paradigm of viral infectious watermarking (VIW) model that uses the biological virus theory for an integrated copyright protection system that copes with various video codecs such as HEVC. The proposed method controls the video quality and the insertion strength. This method hides the watermark into quantized coefficients and also hides the side information for the recovery of the original video content without frame delay.
· The article is well-written and easy to follow. First of all, the authors present the video watermarking for copyright and ownership protection, the compressed DCT domain, and the problematic of the evolution of video codecs and the existed video watermarking systems. Then, they presented the Biological VIW architecture, and the proposed VIW model.
· The proposed method was completely implemented and evaluated by the authors.
· The codec robustness, transparency, capacity and computational time are presented.
· The proposed method has tolerances to attack in a compressed video stream.
· Also, if non-authorized codecs are used to decode watermarked video, the proposed method provides the video with low resolution.
Author Response
Answer : Thanks for your review comments. We modified some parts according to other reviewer. The following is the modified part.
Section 1. Introduction : We present the originality of this manuscript compared to our previous work [20].
Section 2. VIW Architecture : We note the contribution compared to our previous infectious watermarking model.
Section 4. Experimental Results. Subsection 4.1 : We explain more information for experimental environments; test codecs/sequences, computing environment, etc. Subsection 4.2 : We compared BER and PSNR of conventional method and our method.
We colored this modified part blue.
